# Multi-Omics Approaches Uncovered Critical mRNA–miRNA–lncRNA Networks Regulating Multiple Birth Traits in Goat Ovaries

**DOI:** 10.3390/ijms252212466

**Published:** 2024-11-20

**Authors:** Weibing Lv, Ren An, Xinmiao Li, Zengdi Zhang, Wanma Geri, Xianrong Xiong, Shi Yin, Wei Fu, Wei Liu, Yaqiu Lin, Jian Li, Yan Xiong

**Affiliations:** 1College of Animal and Veterinary Sciences, Southwest Minzu University, Chengdu 610041, China; lwb18794850948@163.com (W.L.); junhuaishi@163.com (R.A.); 15833262132@163.com (X.L.); 13998732915@163.com (Z.Z.); 19822961911@163.com (W.G.); xianrongxiong@163.com (X.X.); raulyinshi@163.com (S.Y.); fuwei@swun.edu.cn (W.F.); 22100098@swun.edu.cn (W.L.); linyq1999@163.com (Y.L.); 2Key Laboratory of Qinghai-Tibetan Plateau Animal Genetic Resource Reservation and Utilization, Ministry of Education, Southwest Minzu University, Chengdu 610041, China; 3Key Laboratory of Qinghai-Tibetan Plateau Animal Genetic Resource Reservation and Utilization, Sichuan Province, Southwest Minzu University, Chengdu 610041, China; 4Key Laboratory for Animal Science of National Ethnic Affairs Commission, Southwest Minzu University, Chengdu 610041, China

**Keywords:** Chuanzhong black goat (CBG), Tibetan goat (TG), ovary, multiple-birth traits, ceRNA networks

## Abstract

The goat breeding industry on the Tibetan Plateau faces strong selection pressure to enhance fertility. Consequently, there is an urgent need to develop goat lines with higher fertility and adaptability. The ovary, as a key organ determining reproductive performance, is regulated by a complex transcriptional network involving numerous protein-coding and non-coding genes. However, the molecular mechanisms of the key mRNA–miRNA–lncRNA regulatory network in goat ovaries remain largely unknown. This study focused on the histology and differential mRNA/miRNA/lncRNA between Chuanzhong black goat (CBG, high productivity, multiple births) and Tibetan goat (TG, strong adaptability, single birth) ovaries. Histomorphological analysis showed that the medulla proportion in CBG ovaries was significantly reduced compared to TG. RNA-Seq and small RNA-Seq analysis identified 1218 differentially expressed (DE) mRNAs, 100 DE miRNAs, and 326 DE lncRNAs, which were mainly enriched in ovarian steroidogenesis, oocyte meiosis, biosynthesis of amino acids and protein digestion, and absorption signaling pathways. Additionally, five key mRNA–miRNA–lncRNA interaction networks regulating goat reproductive performance were identified, including *TCL1B*–novel68_mature–ENSCHIT00000010023, *AKAP6*–novel475_mature–ENSCHIT00000003176, *GLI2*–novel68_mature–XR_001919123.1, *ITGB5*–novel65_star–TCONS_00013850, and *VWA2*–novel71_mature–XR_001919911.1. Further analyses showed that these networks mainly affected ovarian function and reproductive performance by regulating biological processes such as germ cell development and oocyte development, which also affected the plateau adaptive capacity of the ovary by participating in the individual immune and metabolic capacities. In conclusion, we identified numerous mRNA–miRNA–lncRNA interaction networks involved in regulating ovarian function and reproductive performance in goats. This discovery offers new insights into the molecular breeding of Tibetan Plateau goats and provides a theoretical foundation for developing new goat lines with high reproductive capacity and strong adaptability to the plateau environment.

## 1. Introduction

The goat (*Capra hircus*) is one of the economically important domestic animals for the development of local ecosystems and livestock economy [1,2]. However, its low lambing rate has become a major factor limiting the sustainable development of this industry [3]. The ovary, as an important organ in female goats, not only secretes reproductive hormones but also responds to hormone action and plays a key role in regulating follicular development, ovulation, and oestrus [4]. Thus, ovarian function would directly reflect the fertility of ewes, making it essential to elucidate its underlying molecular mechanisms [5].

The CBG (Chuanzhong black goat) and TG (Tibetan goat) are notable indigenous goat breeds (*Capra hircus*) in China. CBG is distinguished by rapid growth, substantial body size, superior meat production, high lambing rate (2–3 per litter, averaging 252.00%), and robust reproductive capability [6]. TG, native to the Qinghai-Tibet Plateau, thrives in pastoral zones above 2500 m, exhibiting strong resistance and adaptability to the harsh plateau conditions. Despite TG supplying essential resources for native people with agricultural, economic, and cultural significance, its low reproductive efficiency (one kid annually) significantly impedes the development of this industry [7,8]. Therefore, exploration of the transcriptional regulation networks between CBG and TG would provide the theoretical foundation for goat multiple-birth breeding.

Numerous studies have reported that fertility traits in goats are regulated by multiple major genes, involving ovulation rate and embryo survival. Numerous genes associated with sheep multiple births are identified, such as *BMPR1B* [9], *GDF9* [10,11], and *BMP15* [12]. However, these genes are not entirely applicable to the study of multiple births in goats. Therefore, further exploration of genes related to multiple births in goats is necessary. However, with the rapid development of high-throughput sequencing technologies, there are growing pieces of evidence that non-coding RNAs play important roles in mammalian reproduction. Recent studies have highlighted the wide range of roles of miRNAs in follicular development [13], spermatogenesis [14], hormone secretion [15], and early embryonic development [16]. Also, the published works suggested that lncRNAs regulate early germ cell formation, early embryo implantation and development, corpus luteum formation, and the establishment and maintenance of pregnancy in female animals [17,18].

The goat litter traits are genetically regulated by numerous transcripts. However, analyzing ovary mRNA, miRNA, or lncRNA levels alone does not fully elucidate the underlying molecular genetic mechanism for the multiple-birth trait in goats [19]. Thus, the construction of mRNA–miRNA–lncRNA networks in goat ovaries would give important significance for multiple-birth trait breeding. Here, we focused on analyzing the ovarian tissues of CBG, known for outstanding reproductive performance, and TG, with low reproductive efficiency and characterized by strong adaptability. RNA-seq and small RNA-seq analysis were utilized to deeply explore the key mRNA–miRNA–lncRNA target pairs affecting goat ovarian reproductive activity. This research would provide new insights into understanding the molecular regulatory mechanisms of ovarian development in goats with superior reproductive performance and adaptability, which also offers the theoretical foundation for the local breed germplasm resources.

## 2. Results

### 2.1. Morphological Difference of CBG and TG Ovarian Tissues

H&E-stained paraffin sections of CBG (Figure 1A) and TG ovarian tissues (Figure 1B) revealed morphologically normal and well-developed primordial, primary, secondary, sinusoidal follicles, and corpus luteum. Follicle counting showed more follicles in CBG ovaries than that of TG, though the difference was not statistically significant (*p* > 0.05). The presence of two corpus lutea in the CBG ovaries section suggested that CBGs expelled two mature eggs in a single cycle, consistent with their phenotype of producing multiple lambs (Figure 1A–C). There was no significant difference in the total area of the largest cross-sectional ovarian tissues between CBG and TG (*p* = 0.15) (Figure 1D). However, the medullary area of TG ovaries was significantly larger than that of CBG (*p* < 0.05) (Figure 1E), and the proportion of the medullary area in TG ovaries (60.27% ± 8.14%) was extremely higher than in CBG ovaries (16.02% ± 1.14%) (*p* < 0.01) (Figure 1F). Further analysis of the relative expression levels of marker genes related to ovarian function revealed that the levels of *LHCGR*, *CYP19A1*, *STAR*, *HSD3B7*, and *ESR2* were significantly higher in CBG ovarian tissue compared to TG (*p* < 0.05), while the expression of *FSHR* was significantly lower in CBG ovarian tissue (*p* < 0.05) (Figure 1G). These data indicated that the TG ovary possesses more medullary content and a lower ability of steroidogenesis than those of CBG.

### 2.2. Screening and Functional Enrichment Analysis of DE mRNA in CBG and TG Ovarian Tissues

The ovary from CBG and TG was performed RNA-seq to screen DE mRNA. All the raw data were carried out quality control and clean bases ranged from 14.56 to 16.61 G, Q30 values from 96.67% to 96.94%, and GC content at 50.69% ± 1.15% (Appendix A). Further, >94.22% of reads were mapped to the reference genome, with 5.12%–8.94% mapping to multiple loci and 85.66%–89.26% mapping uniquely (Appendix A). The FPKM density distribution curve indicated high consistency in the expression patterns of protein-coding genes among samples (Figure 2A). The PCA plot showed high clustering within groups and greater inter-group differences than intra-group differences (Figure 2B). Next, a total of 1218 DE mRNAs were identified between the TG and CBG groups using a threshold of *p* < 0.05 and |log_2_FC| > 1 (Figure 2C,D, Appendix A). The histogram (Figure 2C) and volcano plot (Figure 2D) revealed 694 genes significantly up-regulated and 524 genes down-regulated in the CBG compared to the TG. The mRNA expressional trends of 12 randomly selected DE mRNAs measured by RT-qPCR were consistent with sequencing data (Figure 2E), confirming the reliability of the sequencing results for subsequent functional analysis.

To further investigate the biological functions of the DE mRNAs, the GO enrichment analysis showed that 624 DE mRNA were significantly enriched in 329 GO terms among the up-regulated genes in the CBG group (Figure 2F), which predominantly participated in biological functions such as cholesterol biosynthetic process (20 genes, *p* = 1.33 × 10^−22^), cholesterol metabolic process (12 genes, *p* = 4.58 × 10^−8^), and steroid biosynthetic process (9 genes, *p* = 6.44 × 10^−8^). For the molecular function process, this enrichment analysis was enriched in cholesterol transporter activity (6 genes, *p* = 2.73 × 10^−7^), oxidoreductase activity (14 genes, *p* = 9.27 × 10^−5^), and AMP binding (4 genes, *p* = 0.00015) (Figure 2F). Next, 449 genes were significantly enriched in 148 GO terms among the down-regulated genes in the CBG group, which predominantly participate in biological functions such as oocyte differentiation (4 genes, *p* = 3.09 × 10^−6^), meiotic cell cycle (8 genes, *p* = 6.26 × 10^−5^), and multicellular organism development (22 genes, *p* = 0.0003). The enriched molecular function processes included chloride channel activity (8 genes, *p* = 3.66 × 10^−6^) and G protein-coupled peptide receptor activity (5 genes, *p* = 0.0022) (Figure 2G). Moreover, the KEGG enrichment analysis showed that the CBG up-regulated genes were significantly enriched in pathways such as ovarian steroidogenesis (11 genes, *p* = 9.62 × 10^−5^), steroid biosynthesis (12 genes, *p* = 6.63 × 10^−13^), steroid hormone biosynthesis (11 genes, *p* = 0.0004), metabolism of xenobiotics by cytochrome P450 (10 genes, *p* = 0.0006), and PPAR signaling pathway (16 genes, *p* = 9.50 × 10^−8^) (Figure 2H); Among them, *HSD17B7* was significantly up-regulated in CBG ovaries (*p* = 0.00068) and was significantly enriched in steroid biosynthesis (ko00100), steroid hormone biosynthesis (ko00140) and ovarian steroidogenesis ( ko04913). While the down-regulated genes in the CBG were significantly enriched in pathways such as oocyte meiosis (6 genes, *p* = 0.023), hippo signaling pathway (8 genes, *p* = 0.012), cell adhesion molecules (CAMs) (7 genes, *p* = 0.035), and axon guidance (8 genes, *p* = 0.032) (Figure 2I). Collectively, 1218 DE mRNAs were screened between multiple-birth CBG and single-birth TG ovaries, which were mainly associated with steroidogenesis and oocyte meiosis.

### 2.3. Screening and Functional Enrichment Analysis of DE miRNA in CBG and TG Ovarian Tissues

The small RNA sequencing of each CBG and TG ovaries samples yielded raw reads ranging from 21.3 to 22.35 M, with Q20 ranging from 21.02 to 21.75 M and clean reads ranging from 21.00 to 21.73 M after quality control filtering, and with minimum values of 20.36 M (95.64%) and 20.02 M (95.36%), respectively. Clean reads were mapped to the reference genome (Appendix A). A total of 1279 miRNAs were identified, including 414 known and 865 novel miRNAs (Appendix A), of which 91.82% of miRNA lengths were between 21 and 23 nt (Figure 3A). The PCA plot across samples indicated high intra-group clustering with greater inter-group variation than intra-group variation (Figure 3B). Compared to the TG group, the CBG group showed significant upregulation of 65 miRNAs and downregulation of 35 miRNAs, with novel197_mature being the most significantly up-regulated and novel411_mature being the most significantly down-regulated (Figure 3C,D, Appendix A). In addition, the expressional trend of 14 randomly selected DE miRNAs was validated by RT-qPCR (Figure 3E), confirming the reliability of the sequencing results for subsequent analysis.

To further elucidate the biological functions mediated by DE miRNAs, prediction of their target genes and GO or KEGG enrichment analysis were performed. Among the 100 DE miRNAs, 73 DE miRNAs predicted a total of 14,124 target genes, with 21,920 binding sites between the DE miRNAs and their target genes (Appendix A). In the CBG group, the target genes of significantly up-regulated miRNAs were enriched in 1921 GO terms. Specifically, 9455 target genes were enriched in GO-BP such as cell adhesion (367 genes; *p* = 3.67 × 10^−16^), MAPK cascade (181 genes; *p* = 3.77 × 10^−16^), and regulation of sodium ion transmembrane transport (30 genes; *p* = 1.76 × 10^−15^); 10,497 target genes were enriched in GO-CC, including plasma membrane (2402 genes; *p* = 6.02 × 10^−28^) and nuclear bodies (262 genes; *p* = 1.97 × 10^−17^); 9225 target genes were enriched in GO-MF, such as protein serine/threonine kinase activity (398 genes; *p* = 7.32 × 10^−16^) and transcription regulatory region DNA binding (211 genes; *p* = 2.06 × 10^−14^) (Figure 3F). The target genes of significantly down-regulated miRNAs in the CBG group were enriched in 992 GO terms. Specifically, 2067 target genes were enriched in GO-BP, such as endoplasmic reticulum membrane organization (26 genes; *p* = 2.92 × 10^−27^), and regulation of cardiac muscle cell membrane potential (26 genes; *p* = 8.64 × 10^−21^); 2290 target genes were enriched in GO-CC, including locations such as axolemma (24 genes; *p* = 1.44 × 10^−15^) and cytoplasmic vesicles (85 genes; *p* = 4.87 × 10^−11^); 2043 target genes were enriched in GO-MF, such as MAP-kinase scaffold activity (22 genes; *p* = 8.97 × 10^−16^), ATP-dependent protein binding (10 genes; *p* = 2.00 × 10^−10^), and mitogen-activated protein kinase binding (14 genes; *p* = 7.39 × 10^−9^) (Figure 3G).

For the KEGG annotation, the target genes of up-regulated miRNAs in the CBG were enriched in 58 pathways, including the MAPK signaling pathway (295 genes; *p* = 6.87 × 10^−14^), glycerophospholipid metabolism (119 genes; *p* = 5.48 × 10^−9^), phospholipase D signaling pathway (160 genes; *p* = 1.67 × 10^−7^), notch signaling pathway (58 genes; *p* = 1.49 × 10^−6^), tight junction (149 genes; *p* = 2.60 × 10^−6^), and insulin signaling pathway (129 genes; *p* = 0.0002) (Figure 3H). The target genes of down-regulated miRNAs in the CBG were enriched in 63 pathways, notably including inflammatory mediator regulation of TRP channels (48 genes; *p* = 6.12 × 10^−9^), progesterone-mediated oocyte maturation (31 genes; *p* = 0.0001), tight junction (44 genes; *p* = 0.0002), GnRH signaling pathway (33 genes; *p* = 0.0004), T cell receptor signaling pathway (30 genes; *p* = 0.0004), and insulin signaling pathway (40 genes; *p* = 0.0005) (Figure 3I). Altogether, these data revealed 100 DE miRNAs between multiple-birth CBG and single-birth TB ovaries, and their target genes mainly enriched in oocyte maturation-related pathways.

### 2.4. Screening and Functional Enrichment Analysis of DE lncRNA in CBG and TG Ovarian Tissues

After aligning the clean reads to the genome, the transcripts were reassembled using StringTie software (V2.2.3), and a comprehensive screening of candidate lncRNAs was performed using four coding potential analysis methods (CPC, CNCI, Pfam, and PLEK tools). The results showed that 1587 predicted non-coding transcripts were identified by these four software tools (Figure 4A). The FPKM density distribution curve (Figure 4B) demonstrated a non-standard normal distribution with good reproducibility, reflecting a high consistency in lncRNA expression patterns across different samples. The PCA plot of the samples (Figure 4C) showed high clustering within groups, with inter-group differences being greater than intra-group differences. Next, a total of 326 DE lncRNAs were screened, with 236 significantly up-regulated and 90 significantly down-regulated in the CBG group (Figure 4D,E, Appendix A). In addition, 12 DE lncRNAs were randomly selected for RT-qPCR validation and were consistent with the trend of sequencing data (Figure 4F).

To further investigate the biological functions of the 326 DE lncRNAs, we conducted GO and KEGG enrichment analyses, respectively. Further analysis revealed that 181 lncRNAs among all the significantly up-regulated in the CBG group were enriched in 890 GO terms. In the GO-BP category, these genes were primarily enriched in pathways such as negative regulation of cytokinesis (5 genes, *p* = 6.75 × 10^−5^) and embryonic skeletal system morphogenesis (6 genes, *p* = 0.001). In the GO-CC category, these genes were primarily enriched in pathways such as mainly kinetochore (5 genes, *p* = 0.0016) and RNA polymerase II transcription factor complex (4 genes, *p* = 0.0046). In the GO-MF category, these genes were primarily enriched in pathways such as ubiquitin-conjugating enzyme binding (3 genes, *p* = 0.027) and ubiquitin-protein transferase activity (7 genes, *p* = 0.034) (Figure 4G). Among the 90 lncRNAs significantly down-regulated in the CBG group, 64 lncRNAs were significantly enriched in 485 GO terms. In the GO-BP category, these genes were primarily enriched in pathways such as cell proliferation (five genes, *p* = 0.0059) and cell division (three genes, *p* = 0.016). In the GO-CC category, these genes were primarily enriched in pathways such as lysosome (3 genes, *p* = 0.038) and nucleus (24 genes, *p* = 0.013). In the GO-MF category, these genes were primarily enriched in pathways such as DNA binding (nine genes, *p* = 0.047) and ubiquitin-protein transferase activity (three genes, *p* = 0.049) (Figure 4H). The KEGG enrichment analysis of target genes for lncRNAs revealed that among the 236 up-regulated target genes in the TBG group, they were mainly enriched in the Ras signaling pathway (9 genes, *p* = 0.007), amino acid biosynthesis (4 genes, *p* = 0.046), cell cycle (4 genes, *p* = 0.046), and aldosterone-regulated sodium reabsorption (3 genes, *p* = 0.0104) pathways (Figure 4I). The 90 down-regulated target genes in the TBG group were predominantly enriched in protein digestion and absorption (3 genes, *p* = 0.0046), cytokine-cytokine receptor interaction (3 genes, *p* = 0.047), and endocytosis (3 genes, *p* = 0.049) pathways (Figure 4J). These data indicated that 326 DE lncRNAs in CBG and TG ovaries may participate in biochemical processes and hormone synthesis.

### 2.5. Co-Expression Analysis of DE lncRNAs and mRNAs

A Pearson correlation analysis was conducted for the expression of 326 DE lncRNAs and 1218 DE mRNAs, with a threshold of correlation coefficient >0.8 and *p* < 0.05, and 73,181 pairs of co-expressed lncRNA and mRNA were identified (Appendix A). Further GO enrichment analysis was performed on each mRNA co-expressed with lncRNAs, revealing significant enrichment in 1921 GO-BP terms, 299 GO-CC terms, and 743 GO-MF terms. Among these, the genes predominantly participated in biological functions such as cholesterol homeostasis (137 genes) and steroid biosynthetic process (118 genes). The molecular products were involved in AMP binding (120 genes) and cholesterol transporter activity (95 genes) (Figure 5A). KEGG pathway enrichment analysis showed that mRNAs co-expressed with lncRNAs were significantly enriched in 151 pathways, mainly including aldosterone synthesis and secretion (114 genes), ovarian steroidogenesis (94 genes), steroid biosynthesis (132 genes), metabolism of xenobiotics by cytochrome P450 (119 genes), oocyte meiosis (27 genes), and cell cycle (6 genes) (Figure 5B). These data were consistent with the aforementioned DE mRNA enrichment analysis enriched in steroidogenesis and oocyte meiosis in the ovary.

Based on the co-expression results of DE lncRNAs and mRNAs, 134 pairs of lncRNA-mRNA trans-interaction pairs were identified (with at least 10 base pairs of direct interaction and binding free energy not exceeding −100) (Figure 5C). Among these, TCONS_00007977, TCONS_00025045, TCONS_00004033, TCONS_00020587, and TCONS_00010729, respectively, regulate 41, 25, 16, 15 and 11 DE protein-coding genes. Within a range of 100 kb upstream and downstream of each DE lncRNA, all mRNAs significantly co-expressed with the lncRNA (|r| > 0.80 and *p* < 0.05) were identified. The analysis revealed that 20 DE lncRNAs predicted to cis-regulate 18 target genes, forming 20 pairs of lncRNA cis-interactions (Figure 5D). Among these, ENSCHIT00000003463 and ENSCHIT00000003464 co-regulate *PKIB*, while TCONS_00010729 and TCONS_00010733 co-regulate *MYOCD*.

### 2.6. Construction of mRNA–miRNA–lncRNA Regulatory Networks in Goats Ovarian

Based on the expression profiles of 1218 DE mRNAs, 100 DE miRNAs, and 326 DE lncRNAs, the Pearson correlation method was used to calculate the correlations among them, with a threshold of |r| > 0.80 and *p* < 0.05 for identifying interaction pairs. The results of the miRNA–mRNA co-expression analysis, based on the principles of miRNA-mRNA negative correlation expressional pattern and the target binding sequences between miRNA and mRNA, led to the identification of 91 miRNA–mRNA relationship pairs out of 6101 negatively regulated pairs (Figure 6A). Among these, novel568_mature, novel68_mature, novel475_mature, and novel274_mature, respectively, target 22 mRNAs, 19 mRNAs, 12 mRNAs, and 9 mRNAs. The mRNA–lncRNA co-expression analysis based on ceRNA network principles identified 62,771 positively correlated pairs out of 74,023. CeRNA scoring refined these to 56 regulatory pairs (Figure 6B), including ENSCHIT00000003176 targeting 7 mRNAs, ENSCHIT00000010023 targeting 18 mRNAs, and XR_001297206.2 targeting 18 mRNAs et al. miRNA-lncRNA interaction analysis identified 1161 negatively regulated pairs out of 7248, based on miRNA-lncRNA interaction principles. Further binding analysis identified 11 miRNA-lncRNA pairs (Figure 6C), such as chi-miR-324-3p targets 16 lncRNAs, chi-miR-2290 targets 5 lncRNAs, chi-miR-324-5p targets 3 lncRNAs, and chi-miR-93-3p targets 3 lncRNAs, et al.

Next, a total of 67 mRNA–miRNA–lncRNA interaction pairs were identified by integrated DE mRNAs, DE miRNAs, DE lncRNAs, and the ceRNA network (Figure 6D). Notably, novel68_mature targets multiple mRNAs and lncRNAs, including interactions like *TCL1B*-novel68_mature-ENSCHIT00000010023 (r = 0.991), *AKAP6*-novel475_mature-ENSCHIT00000003176 (r = 0.991), *GLI2*-novel68_mature-XR_001919123.1 (r = 0.967), *ITGB5*-novel65_star-TCONS_00013850 (r = 0.979), and *VWA2*-novel71_mature-XR_001919911.1 (r = 0.898), which played a crucial regulatory role in processes such as organogenesis, gametogenesis, steroid hormone secretion, and embryonic development.

Further analysis of the mRNAs involved in the constructed ceRNA pairs revealed that GO enrichment analysis highlighted significant enrichment in pathways such as negative regulation of protein processing (*p* = 0.009), embryonic digestive tract development (*p* = 0.012), and oocyte development (*p* = 0.014) (Figure 6E). KEGG pathway enrichment analysis indicated that these mRNAs were predominantly enriched in pathways such as ECM-receptor interaction (*p* = 0.0002), PI3K-Akt signaling pathway (*p* = 0.0013), and Pantothenate and CoA biosynthesis (*p* = 0.026) (Figure 6F). These data demonstrated extensive inter-regulatory interactions among lncRNAs, miRNAs, and mRNAs, constructing ceRNA interaction networks in goat ovaries and identifying five key ceRNA pairs.

## 3. Discussion

### 3.1. Ovarian Histomorphological Differences Indicate the Medulla Enhancing the Ovarian Adaptability to High Altitudes

The mammalian ovary is a critical component of the female reproductive system, primarily responsible for follicular development, oocyte production, hormone secretion, and regulation of the reproductive cycle, which is essential for reproductive performance and fertility [20]. The functionality of the ovary is dependent on its intact structure and tissue characteristics [21], being composed of an outer cortex and an inner medulla. The cortex primarily contains primary, secondary, and mature follicles, and interstitial cells, serving as the main site for oocyte maturation. The medulla provides nutrients and oxygen for follicular development and acts as a hormone transport conduit, including blood vessels, nerves, and connective tissue [20]. Here, a significantly higher proportion of the medulla in TG than that of CBG was explained by TG experiencing severe nutritional stress with a long seven-month dry grass period in the harsh environment of the Tibetan Plateau [22], which the significant increase in medulla volume in TG to provide more nutrients for the fulfillment of normal follicular development and maintaining reproduction in such a plateau environment.

### 3.2. Numerous DE mRNAs, DE miRNAs, and DE lncRNAs Are Involved in Regulating Ovarian Reproductive Functions and Metabolic Levels

Ovulation is crucial for reproduction, and the intricate transcriptional programs between interstitial cells and granulosa cells, as well as between oocytes and cumulus cells, are essential for the ovulatory response [23]. Our study identified 1218 DE mRNAs, 100 DE miRNAs, and 326 DE lncRNAs between multiple-birth CBG and single-birth TG, suggesting significant differences in the aspects of translation, post-transcriptional regulation, and epigenetic regulation between these two groups. Next, DE mRNAs, DE miRNAs, and DE lncRNAs are all significantly enriched in ovarian steroidogenesis and steroid biosynthesis signaling pathways. Ovarian steroidogenesis primarily synthesizes estrogen and progesterone, vital for female reproductive health and fertility [24]. Steroid biosynthesis involves various steroid hormones, including sex and adrenocorticotropic hormones [25]. The DE mRNAs, miRNAs, and lncRNAs regulated the synthesis of estrogen, progesterone, and other steroid hormones, playing crucial roles in sex hormone production, reproductive cycle regulation, and reproductive health maintenance. Interestingly, DE mRNAs, miRNAs, and lncRNAs were also significantly enriched in the protein digestion and absorption pathway, highlighting the critical role of this process in maintaining nutritional balance within the ovary. Proteins are essential nutrients for animal reproduction, playing a vital role in gonadal development, gamete production, fertilization rates, and embryonic development [26]. Protein intake adequately is crucial for enhancing reproductive efficiency and promoting embryonic development [27]. Thus, the DE mRNAs, DE miRNAs, and DE lncRNAs were involved in reproductive efficiency associated with protein digestion and uptake pathways, which partially contributes to the ovarian adaptation to a high-altitude environment.

### 3.3. Identification of Key mRNA–miRNA–lncRNA Regulatory Networks Governing Lambing Traits and Plateau Adaptation in Goats

The combined analysis of DE mRNAs and DE miRNAs revealed numerous mRNA-miRNA targeting pairs and the steroid hormone biosynthesis pathway were enriched, including *3BHSD*–novel151_mature, *CYP19A1*–chi-miR-184, *HSD17B7*–novel568_mature, and *HSD17B12*–novel82_star. Similarly, enrichments were noted in the aldosterone synthesis and secretion (ko04925) pathway and ovarian steroidogenesis (ko04913) pathway, which was involved in *3BHSD*–novel151_mature, *ADCY8*–chi-miR-502b-3p, *ADCY8*–novel71_mature, and *CYP19A1*–chi-miR-184. Of them, *3BHSD* is a key enzyme catalyzing multiple steps of the steroid hormone (testosterone and estradiol) synthesis pathway, influencing germ cell development, gonadal function, and embryonic implantation and development [28]. *HSD17B7* catalyzed the conversion of androstenedione (A4) to testosterone in androgen synthesis, or estrone (E1) to estradiol (E2) in estrogen synthesis, thereby impacting germ cell development, gonadal function, and reproductive behavior [29]. *HSD17B12* was primarily involved in the metabolism of androgens and estrogens, playing a vital role in the maturation of reproductive cells in mammals [30]. *ADCY8* regulated cAMP levels, influencing the synthesis and release of reproductive hormones [31,32]. *CYP19A1*, as a key enzyme in the conversion of androgens (testosterone) to estrogens (estradiol), was critical for maintaining gonadal function [33]. Thus, these DE miRNAs were identified as key candidate miRNAs regulation on multiple-birth traits through the steroid hormone biosynthesis, aldosterone synthesis and secretion, and ovarian functional pathways in goats.

The ceRNA network represents a sophisticated mechanism of gene expression regulation, involving interactions among various RNA molecules such as miRNAs, lncRNAs, and circRNAs. These RNA molecules compete for binding to the same miRNA response elements (MREs), thereby modulating their degradation rates and translation efficiencies, and consequently influencing intracellular protein-coding gene expression levels [34,35,36]. Here, 67 mRNA–miRNA–lncRNA relationship pairs were identified and enrichment analysis suggested significant regulatory roles in mammalian reproduction, including *TCL1B*–novel68_mature–ENSCHIT00000010023, *AKAP6*–novel475_mature–ENSCHIT00000003176, *GLI2*–novel68_mature–XR_001919123.1, *ITGB5*–novel65_star–TCONS_00013850, and *VWA2*–novel71_mature–XR_001919911.1, et al.

Studies have reported that the *TCL1B* could interact with various proteins, such as Akt (protein kinase B), to regulate immune function, cell metabolism, migration, and apoptosis via the PI3K-Akt signaling pathway [37,38]. *AKAP6*, a scaffold protein, mainly functions in cytoskeletal rearrangement and cell cycle regulation, also modulating the cAMP signaling pathway, thereby influencing metabolism, gene expression, cell proliferation, and differentiation [39,40]. *ITGB5* encodes Integrin beta 5, a cell surface receptor involved in the regulation of the immune system, cell proliferation, differentiation, apoptosis, and metabolism, primarily through the phagosome and PI3K-Akt signaling pathways [38,41,42,43]. *GLI2* was a key component of the Sonic Hedgehog (Shh) signaling pathway and played crucial roles in embryonic development, tissue regeneration, cell proliferation, gonadal development, and germ cell differentiation [44,45]. *VWA2*, containing a VWA domain, was an extracellular matrix protein involved in cell adhesion, proliferation, and tissue remodeling [46]. In summary, the key ceRNA regulatory networks constructed in ovarian tissues might regulate germ cell proliferation, apoptosis, adhesion, migration, metabolic pathways, and the immune system through PI3K-Akt signaling pathway and Hedgehog signaling pathways et al.

The limited biological replication of the selected samples was one of limitations of this study, and the sample size should be increased in future studies. Additionally, this study primarily focused on the reproductive ability and plateau adaptability of goats. In the subsequent breeding programs, we will pay more attention to ensure that the breeding of new lines is scientifically sound through evaluating the feasibility of crossbreeding, conservation and utilization of goat genetic resources.

The aim of this study was to explore the key mRNA–miRNA–lncRNA regulatory networks that regulate the reproductive performance and plateau adaptability of goats. The identified networks could be used as detection tools in the molecular breeding process of goats to screen for eligible CBGs and TGs, which would provide new strategy to produce goat strain with multiple births and high plateau adaptability. It will not only improve the economic benefits for local herders but also promote the sustainable development of plateau pasture ecology.

## 4. Materials and Methods

### 4.1. Experimental Design and Sample Collection

Chuanzhong black goats (CBG) and Tibetan goats (TG) in good health, with similar body weights and consistent ages (4 ± 0.5 years), were selected for this study. A total of 15 CBG ewes with multiple (≥three) consecutive litters, weighing 46.14 ± 4.25 kg, were chosen from pastures in Lezhi County, Ziyang City, and 15 TG ewes with single consecutive litters, weighing 21.85 ± 3.68 kg, were selected from pastures in Yajiang County, Ganzi Tibetan Autonomous Prefecture. All study subjects were managed by traditional grazing in local pastures with free access to grass and water. The selected study subjects were slaughtered by carotid artery bloodletting, a process performed in strict accordance with the Principles for the Ethical Treatment of Southwest Minzu University. Bilateral ovarian tissues were collected from the slaughtered goat after rapid dissection (within 30 min) and washed clean of impurities and blood with PBS (phosphate-buffered saline). The left ovary was immersed in fixative (4% paraformaldehyde) in a 50 mL centrifuge tube for subsequent morphological observation, while the right ovary was placed in cryopreservation tubes, immediately snap-frozen in liquid nitrogen, and brought back to the laboratory to be stored at −80 °C for subsequent total RNA extraction.

### 4.2. Hematoxylin and Eosin (H&E) Staining

The goat ovaries were fixed in 4% paraformaldehyde for 24 h to maintain the structural integrity of the tissue. Then, samples were dehydrated, hyalinized, paraffin-embedded, sectioned, and H&E stained according to the previously described methodological steps, sequentially [47]. The H&E staining process was carried out strictly according to the method steps of Hematoxylin Eosin (H&E) Staining Kit (Cat. No. G1120, Beijing solarbio science & Technology co., Ltd., Beijing, China). Subsequently, images were taken using a Laser Scanning Confocal Microscope (LSM 900, ZEISS, Oberkochen, Germany).

### 4.3. Total RNA Extraction from CBG and TG Ovarian Tissues

Total RNA was extracted from CBG and TG ovarian tissues using the TRIzol reagent method (Invitrogen, Carlsbad, CA, USA) and the procedure was carried out in strict accordance with the instructions of mirVana™ miRNA ISOlation Kit (Ambion-1561, Thermo Fisher, Shanghai, China). Quantitation of total RNA (1.8 < OD260/OD280 < 2.1) was carried out using the Nanodrop 2000 (Thermo Fisher Scientific Inc., Waltham, MA, USA). RNA integrity (RIN ≥7.0, 28S/18S ≥ 0.7) was assessed by Agilent 2100 Bioanalyzer (Agilent Technology, Santa Clara, CA, USA).

### 4.4. Small RNA Sequencing Analysis

#### 4.4.1. Small RNA Library Construction

An amount of 1 μg total RNA of each sample was used for the small RNA library construction using NEB Next Small RNA Library Prep Set for Illumina kit (Cat. No. NEB#E7330S, NEB, Ipswich, MA, USA) following the manufacturer’s recommendations. Briefly, total RNA was ligated to adapters at each end. Then, the adapter-ligated RNA was reverse-transcribed to cDNA and performed PCR amplification. The PCR products ranging from 140–160 bp were isolated and purified as small RNA libraries. Library quality was assessed on the Agilent Bioanalyzer 2100 system. The libraries were finally sequenced using the Illumina Novaseq 6000 platform (Illumina, San Diego, CA, USA), and 150 bp paired-end reads were generated.

#### 4.4.2. Small RNA Bioinformatics Analysis

The obtained raw reads were subjected to quality control, filtering out low-quality reads, resulting in clean reads. The clean reads were compared in the reference genome (Genome Database v.ARS1) to obtain position information on the reference genome (Mapped Reads). Bowtie software (V2.4.4) was used to compare the reads from the reference genome with the Rfam v10.1 database, annotating and filtering out sequences such as rRNA, scRNA, Cis-reg, snRNA, and tRNA. The reads were compared to the reference genome with the Rfam v10.1 database using Bowtie software, the sequences of rRNA, scRNA, Cis-reg, snRNA, tRNA, etc. were annotated, and these sequences were filtered and removed from the Rfam database. Bowtie software was then used to compare and annotate the cDNA sequences, Repbase database and miRBase database (V.22.0) to remove the degraded transcript sequences and repetitive sequences and to realize the identification and annotation of Known miRNAs. Unannotated small RNA sequences were used for novel miRNA prediction using miRDeep2 software (V5.10.1), and the secondary structure of the newly predicted miRNAs was predicted using RNAfold. Expression counts were performed based on the sequences of the identified known mature somatic miRNAs as well as the newly predicted miRNAs.

The *p*-value was calculated using the DEG algorithm in the R package and DE miRNAs were screened with a threshold of *p* -value < 0.05 and |log_2_ FC| > 1. Target gene prediction of DE miRNAs was performed using miranda software (V3.3a) (with parameters: S ≥ 150, ΔG ≤ −30 kcal/mol and Demand strict 5′ seed pairing). The target genes of DE miRNAs were analyzed by GO enrichment and KEGG pathway enrichment and the FDR was obtained by correcting the *p* -value by the Benjamini and Hochberg multiple test.

### 4.5. RNA-Seq Analysis

#### 4.5.1. RNA-Seq Library Construction

For each sample, 1 μg of total RNA was removed rRNA using TruSeq Stranded Total RNA with Ribo-Zero Gold, and then reverse transcribed to synthesize cDNA. The purified cDNA was subjected to end repair, add A-tail, ligate sequencing junctions, and finally amplify. The constructed libraries were quality checked with Agilent 2100 Bioanalyzer (Agilent Technologies, Santa Clara, CA, USA) and then sequenced using Illumina 6000 sequencer (Illumina, San Diego, CA, USA).

#### 4.5.2. Quality Control Filtering and Genome Alignment

After obtaining raw reads from sequencing, residual rRNA sequences were removed using the Sort MeRNA software (V4.3.6). Reads containing adapters, ambiguous bases (N), and low-quality bases were filtered out using the fastp software to obtain high-quality clean reads, which were aligned to the ARS1 genome database (Genome Database v.ARS1) using Hisat2. Subsequently, reads aligned to genes were assembled using the StringTie software, and single transcripts assembled from individual samples were merged into a comprehensive transcriptome.

#### 4.5.3. Differential Analysis of Protein-Coding Genes and lncRNAs

For quantification of protein-coding gene expression, the htseq-count software (V2.0.3) was used, and FPKM values were calculated. StringTie was employed to assemble transcripts, and Cuffcompare was used to compare these transcripts with the reference transcripts individually, determining their positional types. Transcripts longer than 200 bp with at least two exons were selected as candidates for lncRNAs. Their coding potential was evaluated using CPC2, CNCI, PLEK, and Pfam databases. The remaining unannotated transcripts were aligned to known databases using blastn to identify novel lncRNAs. Clean reads were aligned to the transcriptome using Bowtie2, and transcript quantification of lncRNAs was performed using eXpress to obtain FPKM values and counts. Both Bowtie2 and eXpress were run with default parameters. DESeq2 was used for differential expression analysis of protein-coding genes and lncRNAs. DE mRNAs and DE lncRNAs were determined based on a *p*-value < 0.05 and |log_2_ FC| > 1. Further analysis was performed using Pearson correlation analysis based on the expression data of DE lncRNAs and DE mRNAs to calculate the correlation coefficients. A threshold of correlation coefficient > 0.8 and *p*-value < 0.05 was applied to identify co-expression relationships between DE lncRNAs and DE mRNAs.

#### 4.5.4. GO and KEGG Enrichment Analysis of Neighboring Genes of DE mRNAs and lncRNAs

Swiss-Prot (http://www.gpmaw.com/html/swiss-prot.html, accessed on 19 December 2023), an annotated protein sequence database, was used to align mRNA transcript sequences, thereby obtaining Gene Ontology (GO) annotation information for each mRNA transcript. KAAS (http://www.genome.jp/tools/kaas/, accessed on 20 December 2023), provided by the KEGG database, was employed to align mRNA transcript sequences, yielding KEGG annotation information and pathway maps for each transcript. Each GO term or KEGG pathway was considered a functional module, and the significance of differential mRNAs being enriched in GO or KEGG was assessed using a hypergeometric distribution test (*p*-value).

### 4.6. Primer Design and Quantitative Real-Time PCR (RT-qPCR)

#### 4.6.1. Primer Design

Primer sequences for the specific amplification of mRNA, miRNA, and lncRNA were designed using Primer 5.0 (Appendix A) and synthesized at tsingke Biotechnology Co Ltd. (Chengdu, China). We tested the primers to ensure that their amplification efficiency was 90%–110%; amplification curves and agarose gel plots of RCR products determined each primer’s specificity.

#### 4.6.2. RT-qPCR

An equal amount of RNA was taken from each sample and then reverse-transcribed to cDNA using HiScript II Q RT SuperMix for qPCR (+gDNA wiper) (R223, Vazyme, Nanjing, China); then, RT-qPCR was performed using ChamQ SYBR qPCR Master Mix (High ROX Premixed) kit (Q441, Vazyme, Nanjing, China). We used a reaction system with a total volume of 20 µL: cDNA template, 1 µL; Forward primer (10 µM), 1 µL; Reverse primer (10 µM), 1 µL; 2×Plus SYBR real-time PCR mixture, 10 µL; ddH_2_O, 7 µL; The reaction procedures were: 94 °C, 120 s; 94 °C, 15 s; 60 °C, 15 s; 72 °C, 30 s; 45 cycles; the melting curve was machine default setting; the relative expression levels of mRNA and lncRNA were calculated using the 2^−ΔΔCT^ method, with GAPDH serving as the internal control for normalization. The validation of miRNA expression via RT-qPCR was performed as previously described [22].

### 4.7. Data Analysis

Data were quantified using ImageJ Pro Plus (V.6.0) on H&E-stained sections, with measurements taken three times per biological sample and averaged to obtain the raw data. The H&E Staining (*n* = 3), gene expression (*n* = 8), RNA-Seq (*n* = 3), small RNA-Seq (*n* = 3), and RT-qPCR results (*n* = 8) are presented as “MEAN ± SEM”. Statistical analysis was performed using SPSS (V.25) software, employing an independent samples t-test to determine significance, with * *p* < 0.05, ** *p* < 0.01, and *** *p* < 0.001 indicating significant differences.

## 5. Conclusions

The goat breeding industry on the Tibetan Plateau is under strong selection pressure to improve reproductive performance; therefore, there are urgent needs to develop goat lines with higher fecundity and adaptability. Here, 1218 DE mRNAs, 100 DE miRNAs, and 326 DE lncRNAs were identified in CBG and TG ovaries, indicating that a large number of coding and non-coding RNAs are involved in the regulation of ovarian function and plateau adaptation. Additionally, five key mRNA–miRNA–lncRNA interaction networks were identified to regulate goat reproductive performance. Further analyses showed that these networks mainly affected ovarian function and reproductive performance by regulating biological processes such as germ cell development and oocyte development. They also influenced plateau adaptation of the ovary and high-altitude adaptation of the goat by participating in the immune and metabolic capacities of the body. This provides a new perspective for molecular breeding of goats on the Tibetan Plateau and a theoretical basis for breeding new goat lines with high reproductive capacity and high-altitude adaptability (Figure 7). In the future, we will use mRNA–miRNA–lncRNA as genetic markers in goat breeding to more accurately identify individuals with high fertility and high plateau adaptability, thereby accelerating the breeding process.

## Figures and Tables

**Figure 1 ijms-25-12466-f001:**
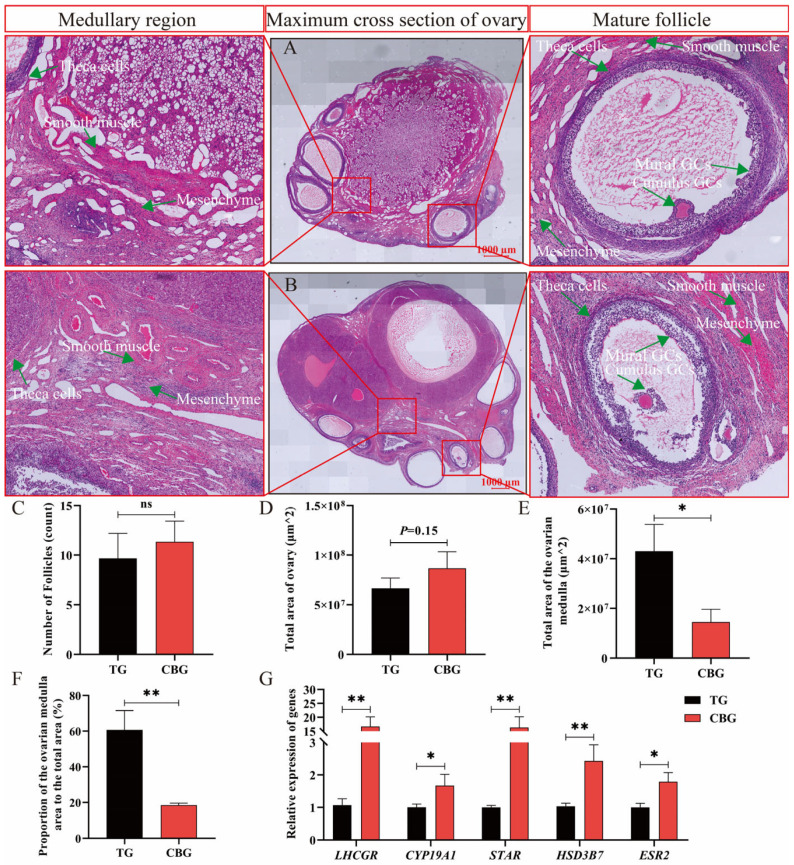
Morphological and statistical analysis of ovarian tissues. (**A**,**B**) H&E staining image of the maximum cross-section of TG (**A**) and CBG (B) ovaries, scale bar = 1000 μm. (**C**) The number of follicles in ovarian tissue by statistical analysis. (**D**) The total area of the maximum cross-section of ovaries. (**E**,**F**) The area (**E**) and proportion (**F**) of medullary in the maximum cross-section of ovaries. (**G**) Relative expression levels of reproduction-related marker genes in ovarian tissues. Note: ** represents *p* < 0.01, * represents *p* < 0.05, and ns represents *p* > 0.05.

**Figure 2 ijms-25-12466-f002:**
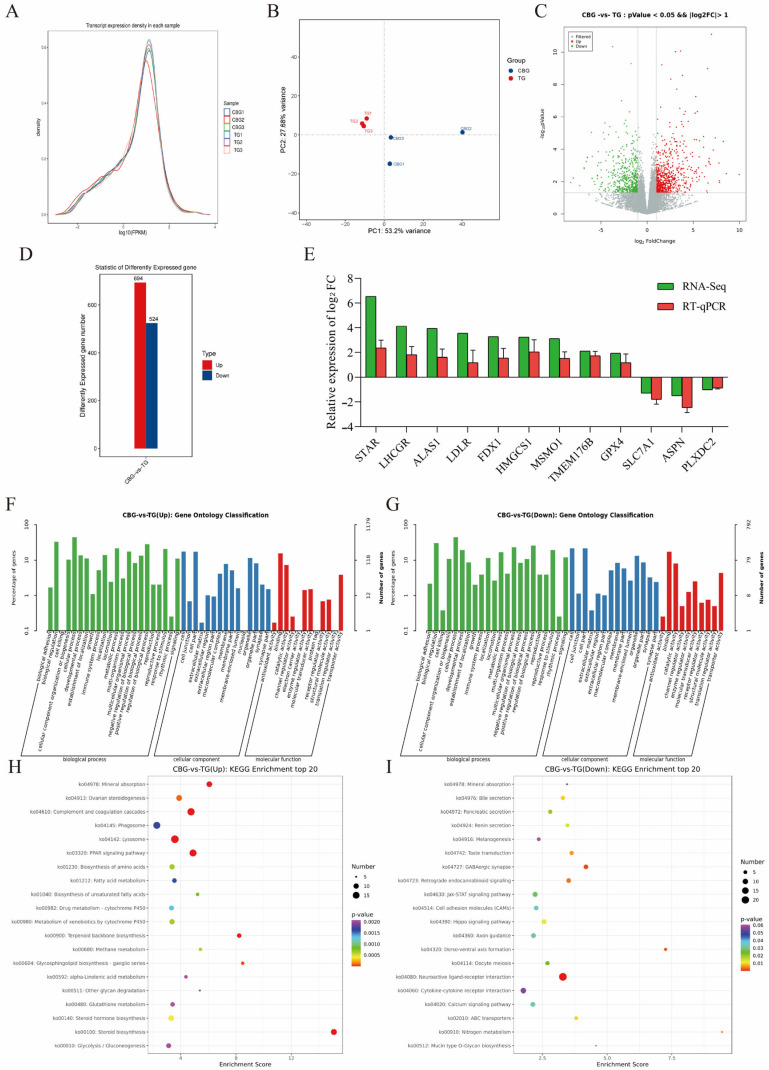
DE mRNA screening and functional enrichment analysis between CBG and TG. (**A**) FPKM density distribution curve. (**B**) PCA plot. (**C**) Volcano plot of DE mRNAs. The *X*-axis was log_2_FoldChange and the *Y*-axis was −log_10_pValue. (**D**) Bar chart of DE mRNAs statistics. The *x*-axis represents the comparison groups, and the y-axis represents the number of differential genes in each group. (**E**) RT-qPCR validation of DE mRNAs. (**F**,**G**) GO enrichment analysis for up-regulated mRNAs (**F**) and (**G**) down-regulated mRNAs. The *x*-axis represented GO term names, and the *y*-axis represented −log_10_pValue. (**H**,**I**) KEGG enrichment analysis for up-regulated mRNAs (**H**) and (**I**) down-regulated mRNAs. The *X*-axis represented the enrichment score.

**Figure 3 ijms-25-12466-f003:**
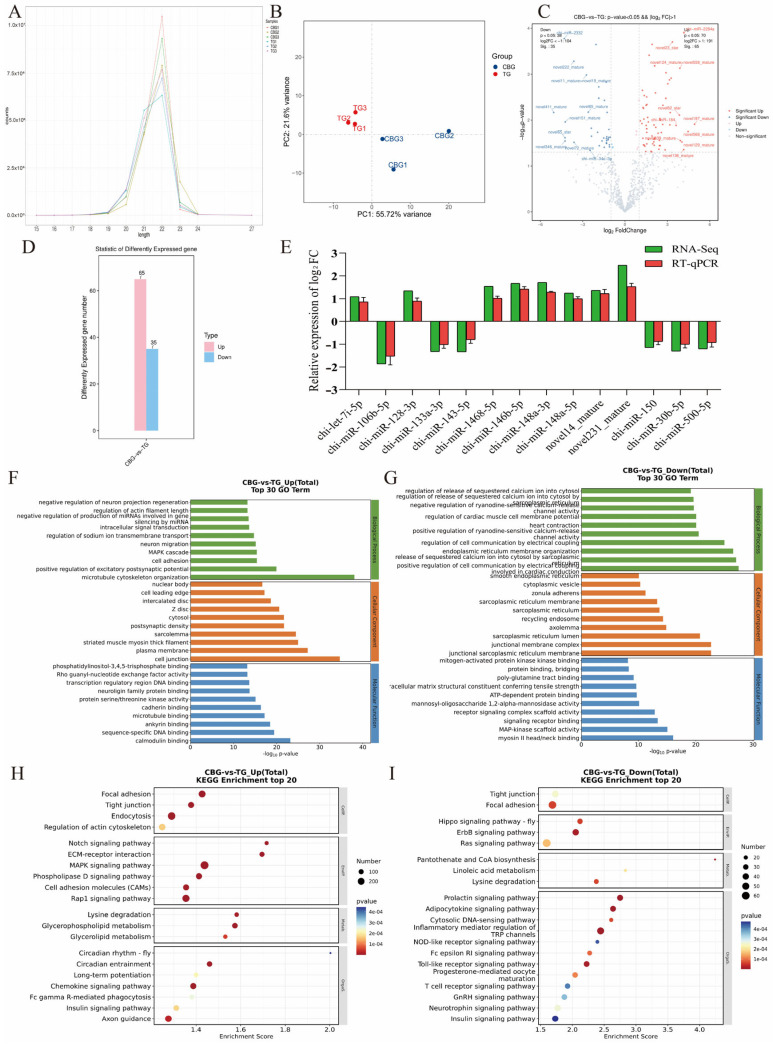
DE miRNA screening and functional enrichment analysis. (**A**) Length distribution of miRNAs. (**B**) PCA plot. (**C**) Volcano plot of DE miRNAs. (**D**) Histogram of DE miRNAs number in these two groups. (**E**) RT-qPCR validation of DE miRNAs. (**F**,**G**) Top 10 bar plots of GO enrichment analysis for target genes of up-regulated miRNAs (**F**) and down-regulated miRNAs (**G**). The *Y*-axis represented GO terms and the *X*-axis was −log_10_pValue. (**H**,**I**) KEGG enrichment analysis for target genes of up-regulated miRNAs (**H**) and down-regulated miRNAs (**I**). The *X*-axis was the enrichment score.

**Figure 4 ijms-25-12466-f004:**
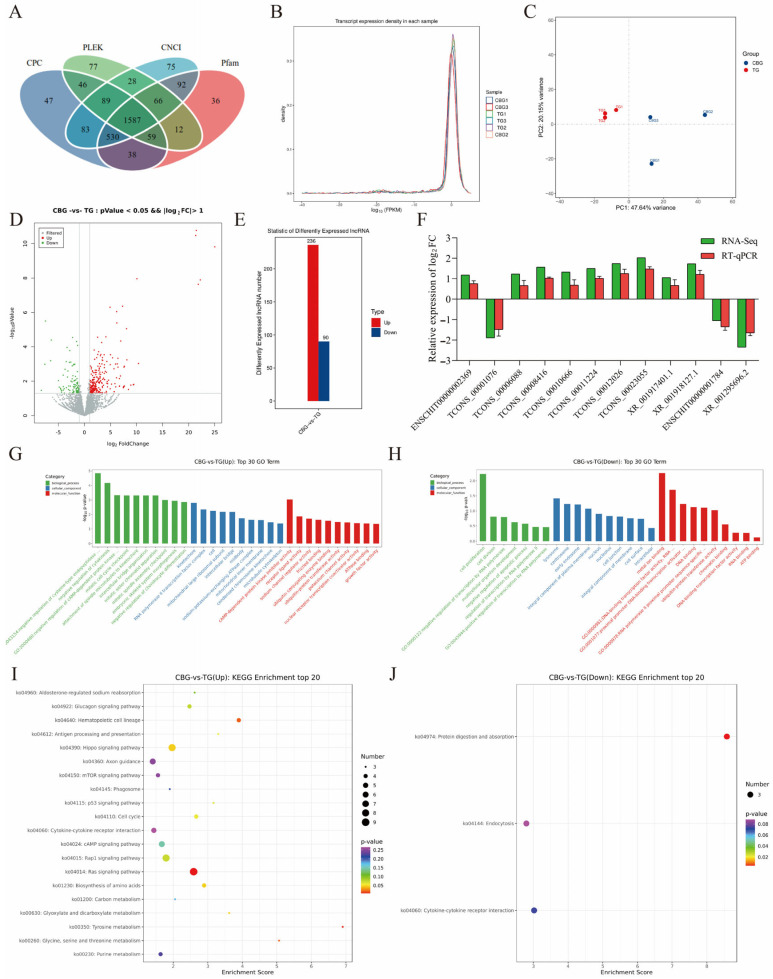
DE lncRNA screening and functional enrichment analysis in the CBG and BG. (**A**) Venn diagram of coding potential prediction results for candidate lncRNAs. (**B**) FPKM density distribution curve. (**C**) PCA plot. (**D**) Volcano plot of DE lncRNAs expression. (**E**) Histogram of DE lncRNAs statistics. (**F**) RT-qPCR validation of DE lncRNAs. (**G**,**H**) GO enrichment analysis for up-regulated lncRNAs (**G**) and down-regulated lncRNAs (**H**). The *X*-axis represented GO term names; The *Y*-axis represented −log_10_pValue. (**I**,**J**) KEGG enrichment analysis for up-regulated lncRNAs (**I**) and down-regulated lncRNAs (**J**). The *X*-axis represented the enrichment score, with larger bubbles indicating more DE lncRNAs, and bubble color ranging between purple, blue, green and red, with smaller *p* values indicating higher significance. G: GO:0043154： negative regulation of cysteine-type endopeptidase activity. GO:2000480： negative regulation of cytokine-mediated signaling pathway. H: GO:0000981： DNA-binding transcription factor activity, RNA polymerase II-specific. GO:0001077： proximal promoter DNA-binding transcription activator activity, RNA polymerase II-specific. GO:0000978： RNA polymerase II cis-regulatory region sequence-specific DNA binding.

**Figure 5 ijms-25-12466-f005:**
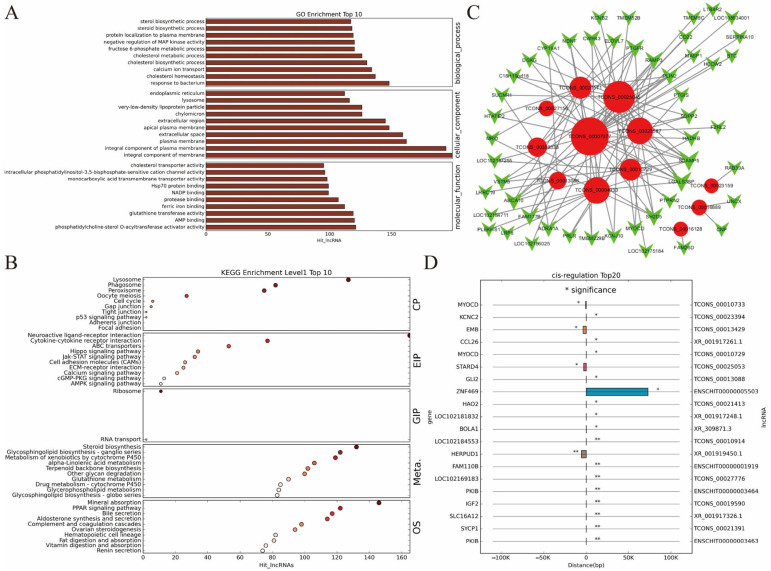
Co-expression analysis of lncRNAs and mRNAs. (**A**) GO enrichment analysis of co-expressed DE genes with lncRNAs (top 10). The *X*-axis represented GO terms, and the *Y*-axis represented the number of lncRNAs enriched. (**B**) KEGG enrichment bubble chart of co-expressed DE genes with lncRNAs (top 10). The *X*-axis represented the enrichment score, with larger bubbles indicating more DE genes in the term, and bubble color ranging from gray to red, reflecting decreasing *p* -values and increasing significance. CP: Cellular Processes; EIP: Environmental Information Processing; GIP: Genetic Information Processing; HD: Human Diseases; Meta.: Metabolism; OS: Organismal Systems. (**C**) Analysis of lncRNA trans-acting target genes. The red circles represented lncRNAs, green inverted triangles represented genes, and node size indicated quantity. (**D**) Analysis of lncRNA cis-acting target genes. The left and right sides of the *y*-axis represented mRNA and lncRNA, respectively. The *x*-axis indicated the distance between mRNA and lncRNA, with negative values indicating upstream and positive values indicating downstream. Identical lncRNAs were represented by the same color bar chart. Note: ** represents *p* < 0.01, * represents *p* < 0.05.

**Figure 6 ijms-25-12466-f006:**
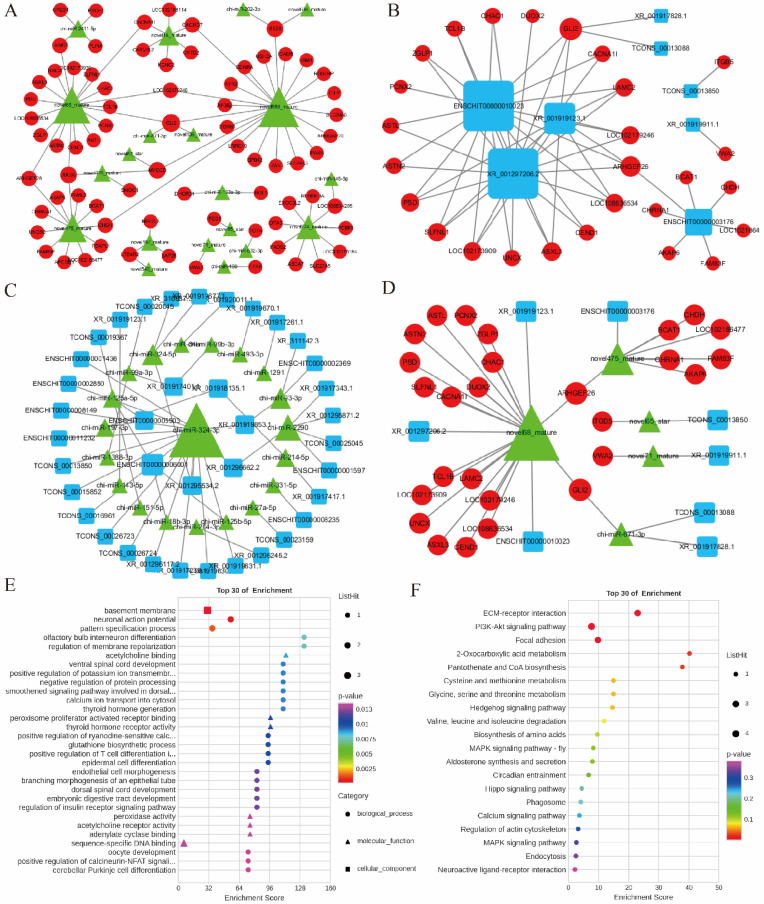
CeRNA interaction network analysis. (**A**) Co-expression network analysis of miRNAs and mRNAs. (**B**) Regulatory network analysis of mRNAs and lncRNAs. (**C**) Regulatory network analysis of miRNAs and lncRNAs. (**D**) Regulatory network analysis of mRNAs, miRNAs, and lncRNAs. Red circles represented mRNAs, green triangles represented miRNAs, and blue rounded rectangles represented lncRNAs. The size of the shapes indicated the quantity. (**E**) GO enrichment analysis of mRNAs in ceRNA. The *Y*-axis represented GO terms, and the *X*-axis represented the enrichment score. (**F**) KEGG enrichment analysis of mRNAs in ceRNA. The *X*-axis represented the enrichment score, and the *Y*-axis represented enriched pathways.

**Figure 7 ijms-25-12466-f007:**
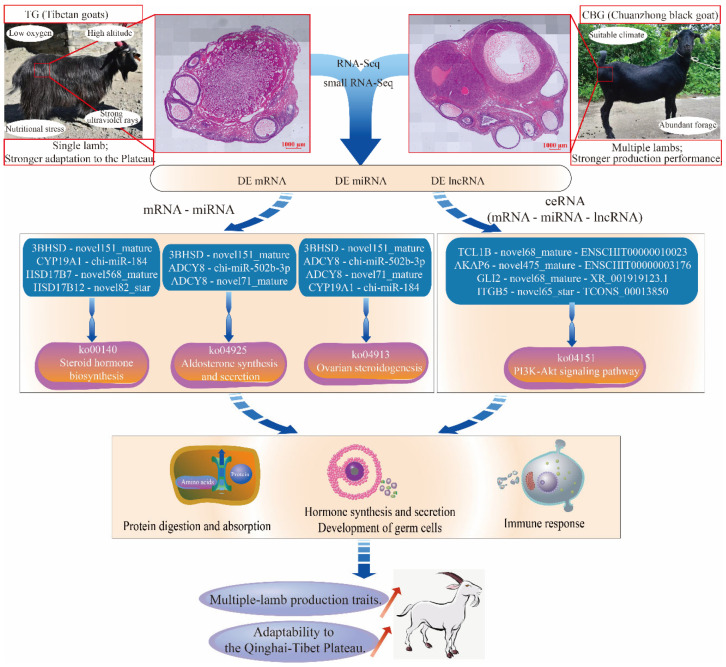
Key mRNA–miRNA–lncRNA regulating goat lambing traits and plateau adaptability.

## Data Availability

The raw data for RNA-Seq and small RNA-Seq in our manuscript have been deposited in the Sequence Read Archive (SRA) of the NCBI, Accession No. SRR25734054–SRR25734059 (small RNA-Seq); SRR28200741- SRR28200746 (RNA-Seq).

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
