# Peer review of "Multi-Omics Approaches Uncovered Critical mRNA–miRNA–lncRNA Networks Regulating Multiple Birth Traits in Goat Ovaries"

_ijms, 2024, doi:10.3390/ijms252212466_

Round 1

Reviewer 1 Report

Comments and Suggestions for Authors

The manuscript titled “Multiple omics reveal key mRNA-miRNA-lncRNA networks regulating multiple-birth trait in goat ovaries” (Manuscript ID: ijms-3241521), mainly compared gene expressions in ovarian of prolific and non-prolific goats, constructing an mRNA-miRNA-lncRNA regulatory network. This research enhances the foundational understanding of goat follicle development and provides a theoretical basis for breeding goats with high fertility and adaptability. However, there are some issues need to be concerned before acceptance for publication.

1. In the Figure 1A, it was interestingly that Tibetan goat had higher proportion of medulla than Chuanzhong black goats. What’s the phyisological significance for this phenomenon, and please discussed this point in the discussion?

2. In the Figure 1G, the relative level of all the steriordgenesis related genes was higher in the CBG than TG, why? Please explain this point.

3. The figures of all the GO/KEGG pathways analysis were not clear and please provide the high resolution images.

4. HSD17B7 plays a crucial role in regulating germ cell development. Although the authors provide a detailed discussion, the description in the results section is not sufficiently clear.

5. The reagents are not labeled as the source. It is recommended to unify them in the manuscript.

6. Does the sample collection comply with animal health and welfare standards? A more detailed operating procedures on collecting the tissue should be provided.

7. Abbreviations should be spelled out in full the first time they appear in the main text to ensure reader comprehension, such as CBD and TG in line 47.

8. Gene names should be italicized. Please check the entire text.

9. The reference formatting needs to be consistent.

10. Writing and grammar errors need to be checked in the manuscript, such as:

a)        Line 45: “ and elucidation molecular mechanism of goat ovary function is necessary” should be “, making it essential to elucidate the molecular mechanisms underlying goat ovary function”.

b)        Line 89: “can” should be “could”.

c)        The term "downregulated" should be "down-regulated." Please review the manuscript for consistency.

d)        Line 278: “of” should be “in”.

Author Response

The manuscript titled “Multiple omics reveal key mRNA-miRNA-lncRNA networks regulating multiple-birth trait in goat ovaries” (Manuscript ID: ijms-3241521), mainly compared gene expressions in ovarian of prolific and non-prolific goats, constructing an mRNA-miRNA-lncRNA regulatory network. This research enhances the foundational understanding of goat follicle development and provides a theoretical basis for breeding goats with high fertility and adaptability. However, there are some issues need to be concerned before acceptance for publication.

Comments 1: In the Figure 1A, it was interestingly that Tibetan goat had higher proportion of medulla than Chuanzhong black goats. What’s the phyisological significance for this phenomenon, and please discussed this point in the discussion?

Response 1: Thank you for pointing this out. We agree with this comment. In mammalian ovaries, the medulla is a central tissue area composed mainly of blood vessels, connective tissue, nerve fibers, and lymphatic vessels. Blood vessels supply blood to the ovary, crucial for follicle development, hormone transport, and metabolite exchange. Connective tissue provides structural support, enhancing mechanical strength and reducing external damage. Nerve fibers influence ovarian function by regulating blood flow, hormone secretion, and follicle development through neural signaling. Lymphatic vessels, a key part of the lymphatic system, help clear cellular metabolites and regulate local immunity, protecting the ovary from infection and damage. The ovarian medulla plays a vital role in structural support, blood supply, neural regulation, and immune defense, ensuring normal ovarian function and reproductive health. Understanding medulla function aids in exploring the pathophysiology of ovarian-related diseases. Further discussion of these results has been added to the discussion section. Lines 337 to 352.

Comments 2: In the Figure 1G, the relative level of all the steroidogenesis related genes was higher in the CBG than TG, why? Please explain this point.

Response 2: Thank you for pointing this out. We agree with this comment. In Figure 1G, compared to TG, the expression of steroid hormone synthesis-related genes (such as CYP19A1, STAR, and HSD3B7) is significantly higher in CBG ovaries. This indicates that CBG goats are more active in hormone synthesis and regulation, potentially affecting their reproductive cycle and sexual maturity. The significant increase in ovarian function-related genes (such as LHCGR and ESR2) suggests that CBG goats have stronger ovarian functions, better supporting follicle growth and maturation. This reflects the superior reproductive efficiency, hormone synthesis, and ovarian function of CBG goats compared to Tibetan goats.

Comments 3: The figures of all the GO/KEGG pathways analysis were not clear and please provide the high resolution images.

Response 3: Thank you for pointing this out. We agree with this comment. We enhanced the quality and resolution of the original images during processing. However, the resolution decreases when they are inserted into a Word document. Therefore, we have uploaded higher-resolution images as attachments.

Comments 4: HSD17B7 plays a crucial role in regulating germ cell development. Although the authors provide a detailed discussion, the description in the results section is not sufficiently clear.

Response 4: Thank you for pointing this out. We agree with this comment. We have added a detailed description of HSD17B7 in the Results section (2.2). Lines 142 to 145.

Comments 5: The reagents are not labeled as the source. It is recommended to unify them in the manuscript.

Response 5: Thank you for pointing this out. We agree with this comment. We have included the details of all reagents and kits in the Materials and Methods section.

Comments 6: Does the sample collection comply with animal health and welfare standards? A more detailed operating procedures on collecting the tissue should be provided.

Response 6: Thank you for pointing this out. We agree with this comment. Ethical approval was granted by the Institutional Animal Ethics Committee of Southwest Minzu University, Chengdu, China, prior to the commencement of the study. The approval number is SMU-202309006. The slaughtering process and sample collection were conducted in strict accordance with the Principles for the Ethical Treatment of Animals at Southwest Minzu University to ensure maximum animal health and welfare standards. Detailed procedures are provided in the Materials and Methods section of the manuscript. Lines 440 to 454.

Comments 7: Abbreviations should be spelled out in full the first time they appear in the main text to ensure reader comprehension, such as CBD and TG in line 47.

Response 7: Thank you for pointing this out. We agree with this comment. It has been thoroughly checked and revised.

Comments 8: Gene names should be italicized. Please check the entire text.

Response 8: Thank you for pointing this out. We agree with this comment. It has been thoroughly checked and revised.

Comments 9: The reference formatting needs to be consistent.

Response 9: Thank you for pointing this out. We agree with this comment. The reference format throughout the text has been modified to strictly adhere to the journal's requirements. Lines 595 to 722.

Comments 10: Writing and grammar errors need to be checked in the manuscript, such as:

  1. a) Line 45: “… and elucidation molecular mechanism of goat ovary function is necessary” should be “…, making it essential to elucidate the molecular mechanisms underlying goat ovary function”.
  2. b) Line 89: “can” should be “could”.
  3. c) The term "downregulated" should be "down-regulated." Please review the manuscript for consistency.
  4. d) Line 278: “of” should be “in”.

Response 10: Thank you for pointing this out. We agree with this comment. All grammatical issues mentioned above have been corrected in the original text. Similar issues have been identified and corrected throughout the document.

Reviewer 2 Report

Comments and Suggestions for Authors

The introduction of the paper is adequate to make the objective of the work clear.

The results and discussion is very extensive and detailed, more work needs to be done on some of the figures (Figure 6 for example) where it is not clear due to the font size used. Regarding the design of the work I have some concerns. The sample size is extremely limited, in these cases it is imperative that the authors present evidence of the power test performed to carry out the experiment. In these circumstances they could elaborate a little on the validity of the results in such a small sample.

There are some additional questions that arise and that there should be prior information on the productive behaviour of these breeds, for example:  Is there any effect of the age of the ewe on the number of offspring at calving in these breeds?

Author Response

Comments 1: The introduction of the paper is adequate to make the objective of the work clear.

Response 1: Thank you for taking the time to review my manuscript and for recognizing our work. We will continue to work hard. Thank you again!

Comments 2: The results and discussion is very extensive and detailed, more work needs to be done on some of the figures (Figure 6 for example) where it is not clear due to the font size used. Regarding the design of the work I have some concerns. The sample size is extremely limited, in these cases it is imperative that the authors present evidence of the power test performed to carry out the experiment. In these circumstances they could elaborate a little on the validity of the results in such a small sample.

Response 2: Thank you for taking your valuable time to review our manuscript, for recognizing our work, and for your valuable comments. First, regarding the issue of unclear fonts in the images (e.g., Figure 6), we have replaced all images with original, higher-quality, and higher-resolution versions, which are uploaded as attachments.

Secondly, prior to conducting this study, we performed extensive preliminary work to confirm its significance and necessity. We observed the histological morphology of the ovaries through H&E sections (n=3) and found that the medullary region of the TG ovaries was significantly larger than that of the CBG ovaries, a common phenomenon. When studying gene expression, we selected ovarian tissues from 8 goats in each group (Figure 1G; n=8) and found good reproducibility within groups, with differences between groups greater than those within groups. Subsequently, we randomly selected 3 samples per group for RNA-Seq and Small RNA-Seq (n=3), and PCA plots showed good reproducibility within groups, with differences between groups greater than those within groups (Figure 2B, Figure 3B, Figure 4C). Additionally, 12 DE mRNAs, 14 DE miRNAs, and 12 DE lncRNAs were randomly selected for validation, and the trends of RT-qPCR results (n=8) were entirely consistent with RNA-Seq results, indicating high reliability of the RNA-Seq data. The subsequent analyses fully support the reliability of the conclusions. Meanwhile, we will incorporate your valuable suggestions into future experimental designs and will increase the sample size to further verify the accuracy of our conclusions.

Comments 3: There are some additional questions that arise and that there should be prior information on the productive behaviour of these breeds, for example:  Is there any effect of the age of the ewe on the number of offspring at calving in these breeds?

Response 3: Thank you for your valuable comments. Our team, including teachers and key members, has extensive experience in the conservation and utilization of genetic resources of animals on the Tibetan Plateau, as well as in reproduction and embryo engineering and livestock product processing. We possess an in-depth understanding of the rearing management, production performance, environmental adaptability, and breed characteristics of the Tibetan Goat (TG) and the Chuanzhong Black Goat (CBG).

The TG and CBG are both excellent local goat breeds. TG is mainly distributed in plateau areas above 3000 meters above sea level, exhibiting strong adaptability to cold, strong ultraviolet rays, and low oxygen, enabling survival and reproduction in harsh environments. In contrast, CBG adapts to various environmental conditions (hills, mountains, and plains) at lower altitudes, characterized by fast growth, high slaughter rate, and tasty meat.

The number of lambs produced by both TG and CBG is closely related to age. TG ewes generally reach sexual maturity around 2 years old and start lambing. Their reproductive ability peaks between 3 and 5 years, with the highest number of lambs, but declines significantly after 6 years. CBG ewes typically reach sexual maturity and start lambing at 1.5-2 years old. Their reproductive peak occurs between 2 and 5 years, with the highest lamb counts and reproductive efficiency, followed by a significant decline after 6 years.

For this study, we selected TG and CBG ewes that had reached sexual maturity, had continuous production records, and were at their reproductive peak. This research aims to provide new insights into the molecular regulatory mechanisms of ovarian development in goats with superior reproductive performance and adaptability, and to offer theoretical foundations for understanding the genetic mechanisms of local breed germplasm resources.

Reviewer 3 Report

Comments and Suggestions for Authors

Comments on the Quality of English Language

Minor grammatical errors are throughout the paper. I have pointed out a few for the authors to look at. The senior authors need to look at the paper and correct some of these small errors.

Author Response

Thank you for taking the time out of your busy schedule to review our manuscript and for your valuable comments, which have greatly improved its quality. They also provide important guidance for our future research. We have uploaded the complete response to the review comments in a document for your perusal.

Round 2

Reviewer 3 Report

Comments and Suggestions for Authors

IJMS-3241521-Peer-Review-v2-Report

Thank you again for sending the revised version of the manuscript. The authors implemented most of my previous recommendations and suggestions, and the manuscript is better now. I have suggested a few things (mainly grammatical issues and spellings) for the authors to address to further improve it.

Title.

The title of any paper should be attractive and must capture the minds of its readers. To achieve this, the authors may wish to paraphrase the title of this paper. Suggestion: Multiple omics approaches uncovered critical mRNA-miRNA-lncRNA networks regulating multiple birth traits in goat ovaries.

Abstract

a)     Line 16: The word ‘in’ is redundant and should be deleted.

b)    Line 18: The word ‘remain’ should rather be ‘remains’.

c)     Line 24: et al. is redundant and should be deleted.

Introduction

a)     Line 39: Correct the spelling of ‘demostic’, and animal should be in the singular form.

b)    Line 45: Remove the preposition ‘of’.

Discussion

a)     Line 334: Remove the phrase ‘follicles, including’ as it is redundant.

b)    Lines 336-337: Rewrite for clarity. Suggestion: The medulla provides nutrients and oxygen for follicular development and acts as a hormone transport conduit, including blood vessels, nerves, and connective tissue [20].

c)     Lines 342-343: Paraphrase this sentence for more clarity. Suggestion: This distinct morphology of the TG's ovary makes it tolerable to the plateau's harsh conditions, thereby maintaining reproduction in such an environment.

d)    Lines 342-343: correct the sentence. Suggestion: Thus, the DE mRNAs, DE miRNAs, and DE lncRNAs were involved in reproductive efficiency associated with protein digestion and uptake pathways, which partially contributes to the ovary's adaptation to a high-altitude environment.

e)     Lines 342-343: The authors should paraphrase this sentence. Suggestion: This study's limitation is the limited biological replication of the selected samples, and the sample size should be increased in future studies to enhance the accuracy of the results.

Materials and Methods

a)     Lines 448-449: Paraphrase for clarity and more understanding. Suggestion: Then, sequentially, samples were dehydrated, hyalinised, paraffin-embedded, sectioned, and H&E stained according to the previously described methodological steps [47].

b)    Lines 531-533: Paraphrase for clarity. Suggestion: We tested the primers to ensure that their amplification efficiency was 90%- 110%; amplification curves and agarose gel plots of RCR products determined each primer's specificity.

Author Response

Thank you again for sending the revised version of the manuscript. The authors implemented most of my previous recommendations and suggestions, and the manuscript is better now. I have suggested a few things (mainly grammatical issues and spellings) for the authors to address to further improve it.

Title

The title of any paper should be attractive and must capture the minds of its readers. To achieve this, the authors may wish to paraphrase the title of this paper. Suggestion: Multiple omics approaches uncovered critical mRNA-miRNA-lncRNA networks regulating multiple birth traits in goat ovaries.

Abstract

  1. a) Line 16: The word 'in' is redundant and should be deleted.
  2. b) Line 18: The word 'remain' should rather be 'remains'.
  3. c) Line 24: et al. is redundant and should be deleted.

Introduction

  1. a) Line 39: Correct the spelling of 'demostic', and animal should be in the singularform.
  2. b) Line 45: Remove the preposition 'of'.

Discussion

  1. a) Line 334: Remove the phrase 'follicles, including' as it is redundant.
  2. b) Lines 336-337: Rewrite for clarity. Suggestion: The medulla provides nutrients and oxygen for follicular development and acts as a hormone transport conduit, including blood vessels, nerves, and connective tissue [20].
  3. c) Lines 342-343: Paraphrase this sentence for more clarity. Suggestion: This distinct morphology of the TG's ovary makes it tolerable to the plateau's harsh conditions, thereby maintaining reproduction in such an environment.
  4. d) Lines 342-343: correct the sentence. Suggestion: Thus, the DE mRNAs, DE miRNAs, and DE lncRNAs were involved in reproductive efficiency associated with protein digestion and uptake pathways, which partially contributes to the ovary's adaptation to a high-altitude environment.
  5. e) Lines 342-343: The authors should paraphrase this sentence. Suggestion: This study's limitation is the limited biological replication of the selected samples, and the sample size should be increased in future studies to enhance the accuracy of the results.

Materials and methods

  1. a) Lines 448-449: Paraphrase for clarity and more understanding. Suggestion: Then, sequentially, samples were dehydrated, hyalinised, paraffin-embedded, sectioned, and H&E stained according to the previously described methodological steps [47].
  2. b) Lines 531-533: Paraphrase for clarity. Suggestion: We tested the primers to ensure that their amplification efficiency was 90%- 110%; amplification curves and agarose gel plots of RCR products determined each primer's specificity.

Response 1: Thank you very much for your advice. We appreciate the significant effort and time you invested in reviewing our manuscript, which has greatly helped improve its quality. We have revised the title to " Multiple omics approaches uncovered critical mRNA-miRNA-lncRNA networks regulating multiple birth traits in goat ovaries" according to your suggestion. Additionally, we have addressed all the grammatical issues you identified, which can be found in the corresponding paragraphs of the manuscript. Thank you again for your hard work.
